# Nanocellulose-Based Composite Materials Used in Drug Delivery Systems

**DOI:** 10.3390/polym14132648

**Published:** 2022-06-29

**Authors:** Ying Huo, Yingying Liu, Mingfeng Xia, Hong Du, Zhaoyun Lin, Bin Li, Hongbin Liu

**Affiliations:** 1Tianjin Key Laboratory of Pulp and Paper, School of Light Industry Science and Engineering, Tianjin University of Science & Technology, Tianjin 300457, China; 17862325433@163.com (Y.H.); xiamingfeng930@163.com (M.X.); duhongxu2021@163.com (H.D.); 2Key Laboratory of Pulp and Paper Science & Technology of Ministry of Education, Qilu University of Technology (Shandong Academy of Sciences), Jinan 250353, China; linzhaoyun123@126.com; 3CAS Key Laboratory of Biofuels, Qingdao Institute of Bioenergy and Bioprocess Technology, Chinese Academy of Sciences, Qingdao 266101, China

**Keywords:** nanocellulose, drug delivery, microparticles, film, hydrogel, aerogel

## Abstract

Nanocellulose has lately emerged as one of the most promising “green” materials due to its unique properties. Nanocellulose can be mainly divided into three types, i.e., cellulose nanocrystals (CNCs), cellulose nanofibrils (CNFs), and bacterial cellulose (BC). With the rapid development of technology, nanocellulose has been designed into multidimensional structures, including 1D (nanofibers, microparticles), 2D (films), and 3D (hydrogels, aerogels) materials. Due to its adaptable surface chemistry, high surface area, biocompatibility, and biodegradability, nanocellulose-based composite materials can be further transformed as drug delivery carriers. Herein, nanocellulose-based composite material used for drug delivery was reviewed. The typical drug release behaviors and the drug release mechanisms of nanocellulose-based composite materials were further summarized, and the potential application of nanocellulose-based composite materials was prospected as well.

## 1. Introduction

Drug delivery systems refer to the advanced technologies used for targeted delivery and/or controlled release of therapeutic drugs [1]. In the past few decades, drug delivery systems have received much attention because they offer potential benefits, such as reducing side effects, improving therapeutic effects, and possibly reducing doses of drugs [2]. There are three key factors required in an effective drug delivery system, including drug carriers, drug-loading ratio, and controlled release rate [3]. A major problem lies in the selection of suitable, natural, nontoxic and inexpensive materials, while the material should maintain good biological activity and fewer side effects. Commercially available poly(lactide-co-glycolide) (PLGA)-based materials are often used for particle drug release formulations [4]. However, due to their large burst release and acidic degradation behaviors, they are often limited to a certain extent. Nowadays, many natural polymers such as cellulose, starch, and glycogen have been extensive studied for drug delivery applications [5,6,7].

Cellulose is one of the main components in natural plants, with good renewability and biodegradability. It is intertwined with lignin and hemicellulose, which helps to maintain good stability and good strength of the plant [8]. With the development of technology, cellulose has attracted great interest in its new form of “nanocellulose” [9]. Nanocellulose can be mainly divided into three categories, including cellulose nanocrystals (CNCs), cellulose nanofibrils (CNFs), and bacterial cellulose (BC) [10]. Among them, CNCs and CNFs are commonly extracted from a variety of plants or algae through a so-called “Top-down” method. Yet, BC is produced by bacteria, which is a so-called “Bottom-up” approach. Nowadays, nanocellulose has attracted extensive attention for the applications in the fields of drug delivery, because of its large specific surface area, good mechanical strength, stiffness, biocompatibility, and renewability [11]. These characteristics enable nanocellulose with good drug loading and binding capacities [12,13].

Recently, nanocellulose-based materials with different types such as single, hybrid, or nanocomposite systems have been fabricated for application in the drug delivery system [14]. As shown in Figure 1, nanocellulose in types of microparticles, films, hydrogels, and aerogels can be utilized as different drug carriers. This review focuses on nanocellulose-based composite materials used in drug delivery systems, which have different dimensions (1D, 2D, 3D). Hydrophilic and hydrophobic drug release behaviors of nanocellulose-based materials are systematically summarized for the first time. The relationships between the structures of nanocellulose-based materials and drug release behaviors are also emphasized. Moreover, the latest research work on nanocellulose-based materials used in drug delivery is introduced in a general overview. The future perspectives with global market value of nanocellulose materials are also systematically summarized.

## 2. Designed Nanocellulose-Based Materials for Drug Delivery

### 2.1. Structures and Characteristics of Nanocellulose-Based Materials

Nanocellulose is a unique and promising natural material derived from native cellulose or bacteria [15,16]. Table 1 summarizes the fundamental physiochemical properties of different types of nanocellulose. The morphology, size, and mechanical properties of different nanocelluloses are dependent on the fibrous raw materials, the isolation methods, and the processing conditions as well as the possible pre- or post-treatments [17,18]. CNFs can be prepared from the pulped form of wood/plants by the methods of combining chemical, enzymatic treatment, and mechanical pressure stratification [19]. CNFs commonly have both amorphous and crystalline regions with diameters ranging from 3 nm to 60 nm [20]. They also have flexible and gel-like consistency properties due to their amorphous regions and micrometer lengths [21]. Moreover, CNCs with elongated crystalline rod-like fragments are commonly obtained by strong acid hydrolysis, especially sulfuric acid hydrolysis [19]. As shown in Table 1, CNCs have shorter lengths and higher crystallinity compared to CNFs [22]. By contrast, BC is synthesized and secreted by a variety of bacteria, such as *Acetobacter, Pseudomonas, and Agrobacterium* [23]. Compared with plant-derived CNFs and CNCs, BC only contains cellulose and does not contain other components such as hemicellulose and lignin [24]. The diameter of BC ranges from 20 to 100 nm, with a high aspect ratio, high crystallinity (84–89%), and good biocompatibility [25] (Table 1). Moreover, BC also has other unique physical properties, such as high degree of polymerization (molecular weight up to 8000 Da), strong water retention capacity (water content up to 99%), plasticity, and hydrophilicity [26]. Therefore, the three-dimensional layered nanostructure of nanocellulose and its physicochemical properties on the nanoscale have opened up new prospects for its application in many fields.

### 2.2. Properties Required for the Nanocellulose-Based Materials Utilized in Drug Delivery

Materials used in drug delivery systems need many required properties, such as good drug-loading capacity, biocompatibility, and biodegradability [27]. The drugs must be released with correct concentrations under a proper rate [28,29]. Nanocellulose-based materials are usually used as a drug delivery matrix and drug excipients [30,31]. Herein, the specific requirements of nanocellulose-based materials used for drug delivery are summarized. These requirements can be divided into physical properties, surface chemistry, and biological properties.

#### 2.2.1. Mechanical Properties

Mechanical properties of nanocellulose (such as Young’s modulus, tensile strength, and toughness) play an important role in sustained drug delivery systems [32]. Because of its disordered amorphous regions and ordered crystalline regions, nanocellulose-based composite materials have good mechanical properties [33,34]. The amorphous regions can contribute to the plasticity and flexibility of nanocellulose-based composite materials [35]. In contrast, the crystalline regions determine the elasticity and stiffness of the materials. Moreover, different types of nanocellulose contain different proportions of amorphous and crystalline domains [34,36]. For example, the stiffness of CNCs is caused by the high crystalline region ratio compared to CNFs and BC [23,37]. Moreover, hydrogen bonding plays an important role in the physical properties of nanocellulose-based composite materials [38]. According to the previous theoretical calculations, as hydrogen bonding is taken into consideration, the longitudinal modulus of cellulose I is about 173 GPa, whereas it reduces to 71 GPa without intramolecular hydrogen bonding [39].

#### 2.2.2. Surface Chemistry

The surface properties of nanocellulose can determine the duration and destination of the prepared drug carriers for the drug delivery [40,41]. Pristine nanocellulose cannot be effectively used as a drug carrier due to its limited water solubility and lack of stability in various buffer solutions [42]. The hydroxyl groups in nanocellulose can offer a broad range of surface functionalization and generate the reactive-charged nanocellulose composites [43,44]. In general, the main objective of the surface modification is to introduce new functional groups into the framework of nanocellulose to attach drugs without altering the morphologies, structures, and crystallinities of nanocellulose-based materials [45,46].

#### 2.2.3. Biocompatibility and Toxicity

Biocompatibility is an essential requirement for biomedical materials [47]. It refers to foreign substances embedded in the body being able to exist consistently with tissues without causing injurious changes [48]. It is reported that CNFs prepared by enzymatic hydrolysis have no cytotoxicity at tested concentrations (~10–1000 μg/mL) [49]. In other CNF fabrication processes, it is exposed that some physical, chemical, and even mechanical changes may affect its cytotoxicity to cells. However, many studies have confirmed that there are no signs of toxicity in pure CNFs [50,51], and other studies have reported low toxicity or no obvious toxicity [52]. The biocompatibility of CNFs may be due to their unique 3D nanofibrous network structure, which supports cell penetration and proliferation [53]. De Loid et al. [54] performed the toxicological analyses of depleted CNCs. In vitro experiments showed that there were no significant changes in serum markers, hematology, or histopathology between the control group and the CNC suspension-fed rats. The experiment suggested that ingested CNCs were basically nontoxic and may be harmless when ingested in small amounts. However, the long-term effects of the materials and their effects in vivo have yet to be revealed [55]. Many polymer-based nanoparticles may have side effects on cells, which may take a long time to observe. BC is reported to be nontoxic and does not show any sign of cytotoxicity in mouse subcutaneous tissue. It is useful in the production of tissue-engineered grafts [56]. Since most toxicity studies are conducted through cell tissue culture, it is impossible to give an accurate image of the compatibility of the selected nanocellulose-based materials [57].

Moreover, the biocompatibility of nanocellulose depends on its structural characteristics, application concentrations, research models, cell type, and exposure times. The uptake of nanocellulose uptake into cells is usually low, which will not induce oxidative stress, and will not produce obvious cytotoxic and genotoxic effects. However, macrophages can internalize rod-shaped CNCs due to their phagocytic function, which can lead to moderate to severe inflammatory response. The response is mainly dependent on the functionalization of CNCs [58]. By introducing different chemical groups on the surface of nanocellulose, the proinflammatory response of nanocellulose can be significantly reduced [59]. Therefore, it is necessary to conduct additional immunological studies on nanocellulose-based materials to better understand its impact on innate and adaptive immunity. However, compared with other materials, nanocellulose-based materials are still preferable because their cytotoxicity is relatively low.

#### 2.2.4. Biodegradability

Besides biocompatibility and nontoxicity, biodegradability is another requirement for materials in biomedical applications [60]. A biodegradable material must be degraded in time that matches the regeneration [61]. However, synthetic biopolymers require high energy and temperature to decompose [62]. Cellulose is a well-known natural polymer with biodegradability. It is generally believed that cellulose does undergo chemical decomposition due to an elevated temperature. One of the main volatile decomposition products is levoglucosan (LGA). This process usually leaves a solid carbon residue whose chemical and physical composition are mostly unknown [63]. Moreover, the nanodimensions of cellulose have not lost their biodegradable nature [64]. However, the biodegradability of nanocellulose in animal and human tissues is not clear, since cells are not able to synthesize cellulases. Nonenzymatic, spontaneous biodegradability of cellulose chains may perhaps explain the slow breakdown of unaltered cellulose within the human body [65]. However, this is admittedly conjecture and it has not been adequately studied [65].

### 2.3. Strategies to Prepare Designed Nanocellulose-Based Materials for Drug Delivery

As discussed above, due to the broad advantages of nanocellulose, including its nanoscale size, high surface area, surface tunable chemistry, good mechanical strength, and biocompatibility, researchers have extensively investigated different nanocellulose-based materials for drug delivery applications [66,67]. Intense efforts have been devoted to improving the properties of nanocellulose-based functional materials to fulfill the demands in drug delivery systems. With the rapid development of technology, nanocellulose has been engineered into multidimensional architectures including 1D, 2D, and 3D, which are further transformed into drug carriers with tailorable structures and properties for different purposes.

#### 2.3.1. Strategies to Prepare 1D Nanocellulose-Based Materials

Commonly, drug carriers should have stable storage of drugs and present controlled-drug-release behavior. Therefore, it should be designed to better control the drug release rate, improve the utilization rate, and reduce the side effects of drugs [68,69]. Nanocellulose-based microspheres have large surface areas and good affinity. They have become a key topic in the research of drug sustained-release systems. Moreover, a variety of methods can be used to modify nanocellulose because of its abundance of surface hydroxyl groups. Surface modification provides a valuable opportunity for controlling the structure–function relationship of nanocellulose [70]. It also can be used to modulate the drug-loading ratio toward nonionized hydrophobic drugs [71]. Ullah et al. [72] fabricated drug-loaded BC microparticles. They found that the drug release rate could be controlled. The microparticles could be tuned for use in many different biomedical applications. Lin et al. [73] used CNCs as chemicals in calcium crosslinked alginate microspheres. The addition of the CNCs stabilized the crosslinked alginate matrix, producing a higher drug encapsulation efficiency of theophylline and obtaining controlled-drug-release behavior.

The emulsion method is widely used to prepare drug nanoparticles. The emulsion system is considered as a promising drug delivery system because of its unique characteristics, such as colloidal stability and easy encapsulation of different compounds [74]. It is commonly emulsified and polymerized in aqueous medium by using high-performance surfactant and emulsion stabilizer [75]. Moreover, the encapsulation of drugs in emulsion also helps to preserve and protect drugs from gastrointestinal hydrolysis and enzymatic degradation. As shown in Figure 2, nanocomposite filaments with diverse functions (drug-loading performance, conductivity, or antibacterial properties) were fabricated by the approach of interfacial colloidal nanoparticle complexation (INC), which was fabricated from oppositely charged colloidal nanocelluloses [76]. Figure 2c depicts the cumulative release curves of doxorubicin hydrochloride (DOX) in different pH media. Obviously, the drug release rate was affected by the pH value of the medium solution. The drug release rate of DOX at physiological pH (pH 7.4) was faster than that under acidic conditions (pH 4.0). The differences in the initial burst release behavior in pH 7.4 and 4.0 were presumably attributed to the different swelling ability of the INC filaments in these aqueous buffers. Low et al. [77] reported a Fe_3_O_4_@cellulose nanocrystals-stabilized Pickering emulsion containing curcumin for magnetically triggered drug release. The drug release rate was increased after the stimulation of the external magnetic field. An MTT (3-(4,5-dimethylthiazol-2-yl)-2,5-diphenyltetrazolium bromide) experiment demonstrated that the curcumin-loaded nanomaterials could effectively inhibit human colon cancer cell growth in the presence of a magnetic field. The results suggested that CNCs-based nanomaterials could be used as a promising colloidal drug carrier.

#### 2.3.2. Strategies to Prepare 2D Nanocellulose-Based Materials

Nanocellulose is an excellent film-forming material with a series of properties such as degradability, good biocompatibility, good permeability, and excellent mechanical properties [74,78]. Taking nanocellulose as a film-forming material and supplemented by other functional components, the composite materials with different types can be prepared, such as drug sustained-release films, hemostatic films, and bone repair films [79]. Therefore, nanocellulose usually appears in the form of blend films or composite films as drug carriers [5]. Many antibiotic, antiviral, and anti-inflammatory drugs can be loaded in the nanocellulose-based composite films [80,81].

However, the drug-loading ratios and the drug release capabilities are the two important factors which should be considered in designing the nanocellulose-based composite films [82]. As shown in Figure 3, Mohanta et al. prepared a layer-by-layer (LBL)-assembled film for drug delivery by using the complementary electrostatic and hydrogen bond interaction between positively charged chitosan and negatively charged CNCs [83]. The films consisted of porous nanofibers and can be loaded with a large amount of doxorubicin. The loaded doxorubicin hydrochloride was released in a sustained manner in a physiological condition mimicked by PBS buffer of pH 7.4 and pH 6.4. Saidi et al. [84] prepared the BC/poly(*N*-methacryloyl glycine) (PMGly) composite films for the controlled delivery of diclofenac. The composite films were prepared by in situ polymerization of methacryloyl glycine monomer within the BC network. Diclofenac drugs were loaded into BC/PMGly films by simply immersing wet composite films in the drug solution. The composite films presented a slower drug release rate (9%) under the pH of 2.1 within 24 h, whereas it showed a much higher release (85%) under the pH of 7.4 in the same duration. Therefore, the composite film had controlled and pH-sensitive drug delivery properties, which have potential to be used in both transdermal drug delivery and oral delivery. Poonguzhali et al. [85] prepared alginate/CNCs composite films for in vitro drug release. The results found that CNCs could improve the swelling and mechanical properties of the composite films. Moreover, the composite films exhibited sustained drug release behaviors. In a recent study, authors incorporated honey in the CNC composite films [86]. The drug release ratio was gradually increased and remained consistent for about 48 h in vitro. Moreover, the drug release behaviors of the composite films followed first-order kinetics. Therefore, nanocellulose films have important research value in the areas of sustained and targeted drug delivery.

#### 2.3.3. Strategies to Prepare 3D Nanocellulose-Based Materials

Nanocellulose-based composite hydrogels are similar to extracellular matrices (ECMs), with highly porous structures, high water retention, good mechanical strength, high specific surface area, and good biocompatibility [87]. The drug molecules can be embedded in the network of the nanocellulose-based composite gels [88]. In the following text, nanocellulose-based composite hydrogels and aerogels used for drug delivery are summarized.

##### Nanocellulose-Based Composite Hydrogels

Generally, nanocellulose-based composite hydrogels are commonly formed by physical crosslinking [89] or chemical crosslinking methods [90]. Crosslinked biomaterials for drug delivery are commonly needed to have biodegradability, biocompatibility, and adjustable physicochemical properties [91,92,93]. Physical crosslinking methods typically include electrostatic/ionic interactions, hydrophobic interactions, and *π*–*π* stacking interactions [94]. Function groups such as hydroxyl groups and carboxyl groups play an important role in the physical crosslinking process [95]. Importantly, the physical crosslinking method can avoid the usage of lethal crosslinking agents which are required in chemical crosslinking methods. Moreover, chemical crosslinking methods typically consist of free radical polymerization, condensation reactions, and aldehyde-mediated reactions [96,97]. Therefore, the abundant active groups such as hydroxyl and carboxyl groups in the nanocellulose backbones make them ideal materials to prepare nanocellulose-based composite hydrogels [98,99].

Different types of hydrogels have distinct morphological structures and functional groups, which can affect the drug diffusion paths during the adsorption and release process [100]. In various types of composite hydrogels, smart responsive hydrogels can be used as biomaterials in continuous and targeted drug delivery systems [101]. When the external environment changes, such as temperature, pH, light, and electric field, the smart responsive hydrogel will shrink or expand as required due to the introduction of hydrogen bonds, complexation, ions, noncovalent interactions, and electrostatic interactions [102]. Therefore, the drug molecules loaded in the smart responsive hydrogels can be released from the hydrogels during the above process. As reported previously, Treesuppharat et al. [103] synthesized the composite hydrogel by copolymerization of BC and gelatin. Due to the uniform shape and the size of the BC chains, the prepared hydrogels had thermal stability and the required mechanical properties. Müller et al. [104] fabricated a BC hydrogel as a carrier for the loading of bovine serum albumin. It was found that lyophilized BC hydrogels had lower bovine serum albumin uptake than that of undried BC hydrogels. The drug was released through diffusion and swelling-controlled processes. Moreover, the researchers continued to use luciferase as a model of protein. They found that the activity of the protein may remain unchanged during the binding and releasing from BC hydrogels. Anin, with other collaborators, prepared a BC/acrylic acid (AA) hydrogel [105]. The water absorption results revealed that the maximum swelling of BC/AA composite hydrogels was achieved at a pH of 7 even after 48 h. When the pH was changed to 10, the equilibrium was attained in 24 h. They further found that the composite hydrogels had both pH and thermoresponsive drug release properties [101]. Liu et al. fabricated a porous polydopamine (MPDA)@graphene oxide (GO)/CNFs composite hydrogel using the physical crosslinking method for controllable drug release [106] (Figure 4). They found that near-infrared (NIR) light irradiation and pH change could accelerate the drug release rate. The pH responsiveness may allow the composite hydrogel to release drugs in the bacteria-infected sites under acidic conditions.

Compared with ordinary hydrogels, injectable hydrogels also have been widely explored as an important part in drug delivery systems. Injectable hydrogels have appeared as promising drug delivery materials because of their properties such as similarity to the ECM, ability to access deep-seated areas, highly porous structure, and capability of enclosing cells within the matrix. A polymer solution with a low viscosity and gel ability after injection can be used to obtain injectable hydrogel. The nucleophilic substitution method can be used to prepare injectable hydrogels for drug delivery and tissue-engineering applications [107,108,109,110]. Injectable hydrogels present a free-flow behavior before injection, but spontaneously change to semisolid hydrogel after gel formation due to the chemical or physical crosslinking reactions. Bertsch et al. prepared the CNC composite injectable hydrogels by salt-induced charge screening (Figure 5). The injectability of the hydrogel was evaluated by the combination of shear rheology and capillary rheology, which showed that the CNC hydrogel was transported through the plug flow in the capillary [111]. The CNC hydrogels were used as drug carriers for the in vitro release of bovine serum albumin (BSA), tetracycline (TC), and doxorubicin (DOX) in normal saline and simulated gastric juice. For TC, a burst release was observed within 2 days, whereas BSA and DOX both showed a sustained release for 2 weeks. The different release behaviors were attributed to drug size, solubility, and specific drug–CNCs interactions. Orasugh et al. [112] added CNCs in the triblock poloxamer copolymer (PM) to obtain composite injectable hydrogels. They found that the PM/CNC hydrogel had good strength and a lower gel temperature, which was attributed to the formation of intermolecular hydrogen bonds between the free hydroxyl group of the CNC molecule and PM molecule. Moreover, injectable hydrogels can undergo reversible phase transitions triggered by pH, temperature, solvent composition, electric field, ionic strength, or light [113].

Magnetic hydrogel can keep the drug under the electromotive force for a long time [114]. Commonly, the magnetic drug carrier improves the electromotive force of the affected pathological site, and the drug release behavior is controlled by electromotive force [115,116]. Magnetic drug-loaded hydrogels can improve the drug-loading efficacy, reduce the drug dosage, and reduce the damage to normal organs. Supramaniam et al. [117] synthesized magnetic CNCs (m-CNCs) for the loading of ibuprofen. Then, the composite was merged with alginate to fabricate composite hydrogels. The burst release amount (45–60%) was observed from 0 to 30 min. A sustained drug release period was observed from 30 to 330 min.

##### Nanocellulose-Based Composite Aerogels

Aerogels are commonly referred to as the porous material with high porosity and low density, which are suitable for the storage of various drugs [118]. Aerogels with adjustable pore size and large pore volume can prevent rapid drug release in undesired areas [119,120]. Nanocellulose is a suitable building block to form aerogels, because of its combined light weight and toughness [121]. Moreover, nanocellulose-based aerogels or their derivatives are unique among solid materials due to their low density, high porosity, and good biocompatibility [122]. In brief, the fabrication process of nanocellulose-based aerogels begins with the formation of hydrogels through the chemical or physical crosslinking method [123]. Additionally, then, the hydrogels are converted into aerogel by different drying techniques such as evaporation, supercritical drying, and the freeze-drying method [124]. Before the drying process, the physical and chemical crosslinking processes are crucial to control the formation of the three-dimensional network and the properties of the porous material [125,126]. Compared with the materials obtained by the physical crosslinking method, chemical crosslinking materials have better mechanical stiffness and structural stability [127]. As shown in Figure 6, Chen et al. fabricated a TEMPO-mediated BC composite aerogel with polyethyleneimine (PEI) as a crosslinker [128]. Aspirin was loaded into the composite aerogel by simple adsorption. The drug release behaviors were investigated in the simulated intestinal fluid (SIF) solutions. The accumulative release ratio of aspirin from composite aerogel was 80.6% in SIF condition under pH of 7.5.

Moreover, the multifunctional nanocellulose-based aerogel can be prepared by grafting functional groups onto cellulose nanofibrils. Controlled-drug-release behaviors can be obtained from different pH/temperature/light conditions [129]. For example, Zhao et al. [130] grafted polyethyleneimine on the surface of CNFs to obtain a composite aerogel used for drug delivery. The composite aerogels had a high drug load capacity (287.39 mg g^−1^). Due to the amounts of amine group contained in the polyethyleneimine molecule, the obtained composite aerogels presented good temperature and pH-responsive drug release properties.

## 3. Drug Release Behaviors of Nanocellulose-Based Materials

Drug release behavior is one of the most important properties for drug carriers [131]. The specific surface area of drug carriers plays an important role in controlling the drug-releasing rate [132]. Drug carriers can be used to load hydrophilic or hydrophobic drugs. The hydrophilicity and hydrophobicity of drug carriers not only determine the absorption, distribution, metabolism, and excretion of drugs in vivo, but they also directly affect the value of drugs [133]. For hydrophilic drug carriers, the drug release rate is commonly fast because the surface tension of hydrophilic drug carriers is easy to collapse in aqueous solution. In contrast, the drug release rate of hydrophobic drug carriers is relatively slow [134]. As the structure of hydrophobic materials is more stable in water, the drug release rate of hydrophobic drugs is mainly affected by drug diffusion [135]. In this part, hydrophobic and hydrophilic drug release behaviors of nanocellulose-based materials are summarized.

### 3.1. Hydrophilic-Drug Release Behaviors

Hydrophilic drugs refer to the drugs that can be dissolved in water. Nanocellulose-based nanocarriers exhibit negative interface charges and high specific surface area, which make them suitable as hydrophilic drug carriers [136]. However, due to their limited water solubility, water sensitivity, and lack of stability in various buffer solutions, the original nanocellulose cannot be effectively used as a drug carrier [35]. Therefore, various methods, such as pretreatment and surface modification, have been developed to overcome these limitations and improve specific properties [137]. Commonly, hydrophilic drugs loaded in the nanocellulose-based carriers via electrostatic attractions or covalent-binding reactions [95]. In addition, hydrophilic drugs usually have problems of low intracellular absorption, enzyme degradation, rapid clearance, poor distribution, drug resistance, poor pharmacokinetics, and low treatment index [138]. Therefore, lower encapsulation efficiency and rapid-release behaviors of hydrophilic drugs in nanocellulose-based carriers are the main problems. To overcome the above problems, many strategies have been developed. Generally, hydrophilic molecules can be first loaded into nanospheres; then, the drug-loaded nanospheres are added in the structure of nanocellulose-based hydrogel materials [106]. Table 2 summarizes the nanocellulose-based materials used in hydrophilic-drug delivery applications. Besides delivering the small drug molecule as listed in Table 2, nanocellulose-based materials can be used to deliver proteins and nucleic acids, because nanocellulose-based materials can meet the strict medical requirements of appropriate carriers for protein and nucleic acid fixation. Basu et al. developed calcium-chloride-crosslinked CNF hydrogels for transporting biomolecules [139]. Bovine serum albumin protein was loaded into the hydrogel by the simple immersion method. The large positively charged proteins promote the sustained drug release behavior of CNFs. The electrostatic interaction between the protein and hydrogel was the main factor to promote the physical adsorption of hydrogel structure stability and activity. Therefore, calcium-crosslinked CNFs hydrogels can transport proteins without affecting their activity.

In vitro drug release study is the basic evaluation experiments to determine the suitability of nanocellulose-based drug delivery materials, which are carried out in either phosphate-buffered saline or mimicking-release medium [151]. Cellulose-based sheet materials with antibacterial and wound-healing properties have been reported [152]. Liu et al. [153] reported a nanocellulose-based hydrogel with a packaging structure for on-demand drug delivery. Zeolite imidazolate skeleton-8 (ZIF-8) was grown on the surface of polydopamine (PDA) to obtain PDA@ZIF-8 nanocomposites. Then, PDA@ZIF-8 nanocomposites were added to the CNF network to prepare PDA@ZIF-8/CNF composite hydrogels. Slight burst release behaviors were observed at the beginning. The composite hydrogel also presented a pH-dependent drug release behavior. Moreover, NIR light irradiation can accelerate the drug delivery rate.

In order to effectively adsorb drugs onto the surface of nanocellulose drug carriers, the physicochemical properties of drugs and nanocellulose materials should be considered. Bhandari et al. [149] found that CNF aerogel was suitable for loading water-soluble bendamustine hydrochloride drugs because of the physical adsorption properties of CNF aerogel. The drug-loading ratio of CNF aerogel was 18.98%. The drug release amount was different in different pH conditions. Approximately 69.2% of the drug was released in 24 h at the pH of 1.2, whereas 78% of the drug was released at the pH of 7.4 within 7.5 h. As shown in Figure 7, Li and coworkers [154] synthesized CNF/gelatin aerogels through the chemical-crosslinking method. The composite aerogels were further used as carriers for the controlled released of 5-fluorouracil (5-FU). The 5-FU powder was added into the CNF solution before hydrogel formation, and the maximum encapsulation efficiency of the drug was about 40%. The cumulative release of composite aerogels was close to 100% after 12 h. Moreover, the controlled and continuous release of drugs was realized, as the drugs should be dissolved from the carrier matrix and then diffused through the structure of the network. Cacicedo et al. [155] mixed lipid nanoparticles loaded with doxorubicin into BC composite hydrogels. They found that the drug encapsulation efficiency of neutral doxorubicin was two times bigger than that of cationic doxorubicin. Moreover, the composite hydrogel showed a sustained drug release behavior toward neutral doxorubicin. In vivo analysis showed that the growth of tumors and metastatic events were decreased by a combination of these two drugs.

### 3.2. Hydrophobic-Drug Release Behaviors

Hydrophobic drugs refer to the substances that are less water-soluble but soluble in organic solvents [156]. It is estimated that about 40% of marketed drugs and 60% of compounds in development status are poorly water-soluble [157]. The lower drug-loading ratio is one of the major challenges which hinders their practical applications [158]. As the presence of reactive functional groups on the nanocellulose backbones, a wide range of surface modifications have been carried out to improve their binding properties toward hydrophobic drugs. Table 3 summarizes the nanocellulose-based materials used in hydrophobic-drug delivery applications.

Due to the problems of poor solubility, poor dispersion, lack of uniformity, low bioavailability, and lack of stability, the therapeutic effect of many hydrophobic drugs is limited [166]. However, when drugs are uploaded into the hydrogel system, these defects can be improved to some extent, resulting in solubilization, slow release or controlled release effects, and enhanced stability and biological activity. As shown in Figure 8, Ma et al. added CNCs loaded with curcumin into tara gum/polyvinyl alcohol blend membrane to prepare antioxidant and antibacterial membranes [167]. The release test revealed that curcumin was initially released rapidly into 50% ethanol solution and then released more slowly into the bulk. Moreover, higher temperature could accelerate the release of curcumin. Liu et al. [165] used the one-pot method to grow ZIF-8 and curcumin (Cur) on the surface of PDA-modified CNFs composite hydrogel (PCNFs) to form ZIF-8@PCNFs-Cur composite hydrogel. The loading of ZIF-8 nanoparticles in the PCNFs composite hydrogel structure was expected to increase the loading rate of hydrophobic drugs and prolong the drug release time. The maximum drug encapsulation efficiency and drug-loading ratio of the composite hydrogel were 82 wt% and 4.5 wt%, respectively. It was found that the composite hydrogel had good mechanical properties and sustained drug release properties. In addition, lower pH condition and near-infrared light irradiation could accelerate the drug release behavior. The maximum drug release time of ZIF-8@PCNFs composite hydrogel was 107 h. The mechanism was confirmed as abnormal transport. Plappert et al. [168] found that the surface charge density and carboxylate content in CNF membranes can increase the adsorption ratio of hydrophobic piroxicam. In vitro drug release time was prolonged under simulated human skin conditions. Their findings confirmed that nanofibrous membranes can be potentially used as transdermal drug delivery patches.

Targeted drug delivery and controlled drug release rate are attractive methods to avoid the sudden drug release behaviors [169]. Anirudhan et al. [170] prepared a nanocellulose-based carrier for effective delivery of curcumin. Under the pH of 8.0, the drug-loading ratio was as high as 89.2%. At acidic pH conditions, almost 91.0% of the drug was released within 48 h. Due to the protonation of imine, carboxyl, and hydroxyl groups in the material, the curcumin release amount at pH 5.5 was higher than that at pH 1.2 and 7.4. The electron repulsion may cause expansion in the material and promote the release of curcumin [171]. Luo et al. [172] synthesized BC/graphene oxide composites via a biosynthesis technique as a novel drug delivery carrier for ibuprofen release. The results showed that the presence of graphene oxide promoted the drug-loading ratio and prolonged the sustained release time. The reasons may be related to the increased specific surface area. In vitro drug release studies indicated that the drug release behavior of the composite material followed a non-Fickian diffusion mechanism.

## 4. Drug Release Mechanisms and Mathematical Models of Nanocellulose-Based Materials

Generally, drug release refers to the process in which drug molecules are transferred from the interior to the outer surface of the drug carriers, and finally released into the surrounding environment [173]. The process is governed by the random motion of drug molecules, driven by chemical potential gradients and convection created by osmotic pressure [174]. For the drug carriers with degradability, the drug release rate is mainly controlled by the diffusion of the carrier’s network. As for nondegradable drug carriers, diffusion is the main driving force for drug release. As for nanocellulose-based carriers, drug release can be predicted by the diffusion rate of the carrier [175]. Moreover, the interactions between drug models and nanocellulose backbones as well as the drug solubility in dissolution medium are the main reasons for the determination of the drug release kinetics. As is shown in Table 2 and Table 3, the sustained drug release mechanisms involved in different drug carriers are different. Kolakovic et al. [176] fabricated CNF films with different drugs including indomethacin, itraconazole, and beclomethasone for the sustained drug release applications. They found that the model drugs could be released from the carriers continuously at an interval of three months. The tight fiber network formed by CNFs around drug particles protected them from the impact of the liquid medium environment. The CNF network is also used to maintain the dissolution of drug molecules and create obstacles for the drug’s diffusion. However, the drug release kinetics of different drug models were different. The release kinetics of the indomethacin drug was diffusion-limiting kinetics. However, the release kinetics of itraconazole and beclomethasone were zero-order release. Considering the slower release rate of drugs, the system can be used in parenteral (implant), local (transdermal patch), or ocular applications.

Mathematical models of drug release are aimed to predict drug release rates and drug diffusion behaviors [177]. It helps to optimize drug release kinetics and determine the physical mechanisms by comparing experimental data with mathematical models [178]. Mathematical models can explain the effects of various parameters on the drug release rate, such as the size, shape, and composition of the drug release system. After fitting by a mathematical model, identified drug release behaviors can be used to predict more effective drug formulations and accurate dosing plans [179]. At present, a series of mathematical models such as the zero-order model (Equation (1)), the first-order model (Equation (2)), the Higuchi model (Equation (3)), and the Korsmeyer–Peppas model (Equation (4)) have been proposed and used to predict and explain drug release behaviors [148,180]. However, these models do not limit the shapes, properties, and structures of different drug carriers. The errors may exist in the simulation of drug release behavior [181]. Therefore, more research is needed to reveal the mechanism of nanocellulose-based drug carriers to achieve better drug release behaviors, which will be one of the future research directions.
(1)$$\frac{M_{t}}{M_{\infty }}=K_{0}t$$
(2)$$ln(1-\frac{M_{t}}{M_{\infty }})=-K_{1}t$$
(3)$$\frac{M_{t}}{M_{\infty }}=K_{2}t^{\frac{1}{2}}$$
(4)$$\frac{M_{t}}{M_{\infty }}=K_{3}t^{n}$$

In all the above models, *M*_∞_ and *M_t_* are the cumulative drug release amount at time *t* and infinite time. $\frac{M_{t}}{M_{\infty }}$ is the cumulative drug release fraction. *n* is the drug release exponents from the drug release mechanism (Table 4). *K*_0_, *K*_1_, *K*_2_, and *K*_3_ are the drug release rate constants of Equations (1)–(4), respectively.

## 5. Conclusions and Perspectives

Nanocellulose has been proved to be one of the most prominent green materials in various applications, and it has attracted great interest in academic research and industrial applications, as evidenced by more than 4500 relevant patents and commercial products around the world. This review comprehensively introduced the latest research progress of different nanocellulose-based composites in drug delivery. The unique properties of the materials used in drug delivery systems are discussed systematically. As a natural material, nanocellulose has the potential to be involved in future medical applications. The current research is still at the scientific research stage, and we need to focus on in vivo drug release research on animals in the future. More long-term studies are needed to analyze and assess the potential effects of nanocellulose on humans. Nonetheless, it is difficult for the human body to degrade nanocellulose-based materials, and the interaction mechanism between nanocellulose and cells is still unclear. In the future, it is necessary to explore whether the introduction of nanocellulose is potentially harmful to the skin, such as causing skin hyperplasia, scars, or other complications.

From a scientific and economic point of view, nanocellulose is a resource and gift provided by nature. Driven by the recent extraordinary activities in the field of biomedical applications, nanocellulose will make a breakthrough in drug delivery systems. Moreover, with the continuous optimization and commercialization of nanocellulose, nanocellulose-based composite materials with better-designed structures and multifunctions (e.g., pH or NIR responses) will be undoubtedly extensively used in the field of drug delivery. Despite nanocellulose-based composites having already shown great promise in biomedical fields, the large-scale commercial applications of nanocellulose are closely related to the structure and performance of the materials. Thus, we need commit to solving the current difficulties and challenges. Therefore, with the continuous optimization of nanocellulose production, modification, industrialization, and commercialization, nanocellulose will undoubtedly shine in more fields.

## Figures and Tables

**Figure 1 polymers-14-02648-f001:**
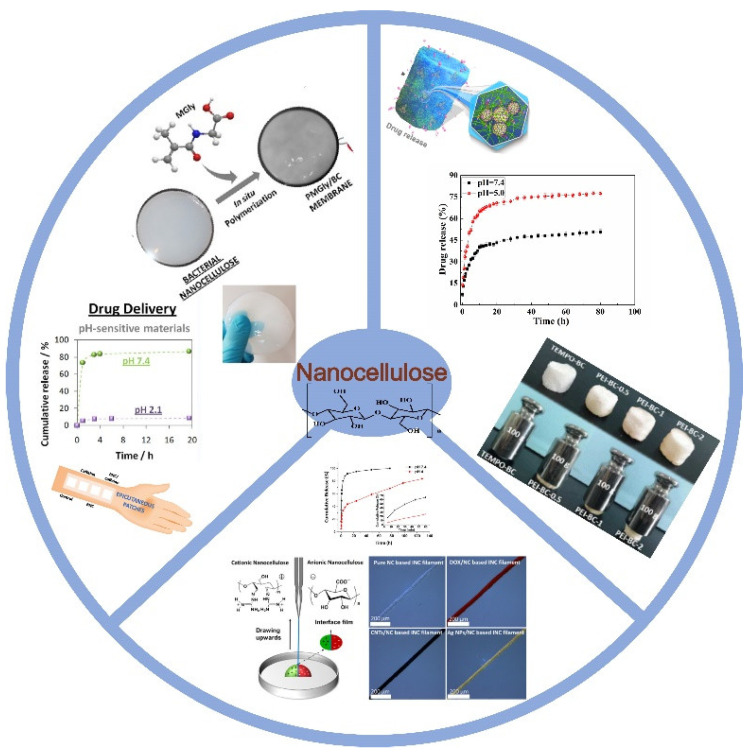
Schematic illustration of different nanocellulose-based materials used for drug delivery.

**Figure 2 polymers-14-02648-f002:**
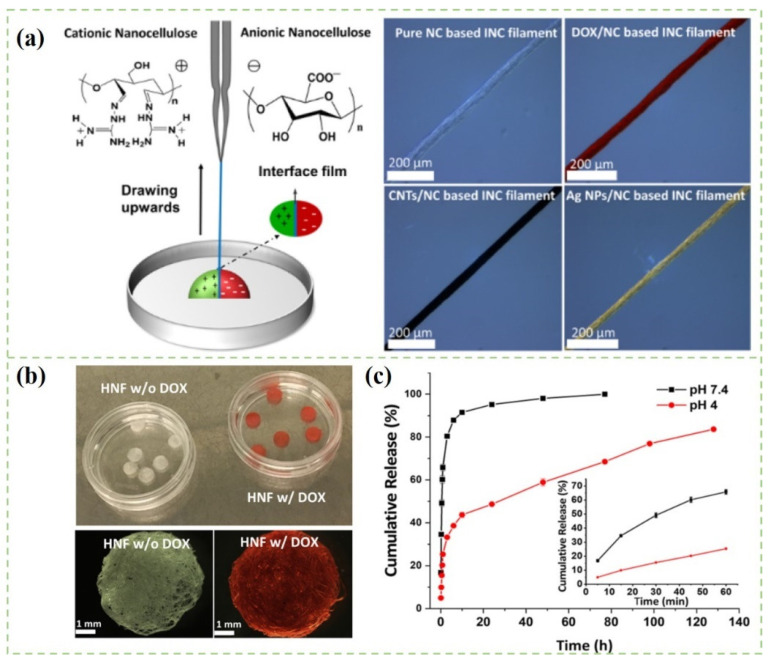
(**a**) Schematic illustration of nanocomposite filament; (**b**) Photography and optical microscopy images of hydroentangled CNCs-based INC filament; (**c**) Cumulative release profiles of DOX-loaded CNCs-based INC filaments in different pH values at 37 °C. The inset shows the cumulative drug release in the first 1 h [76].

**Figure 3 polymers-14-02648-f003:**
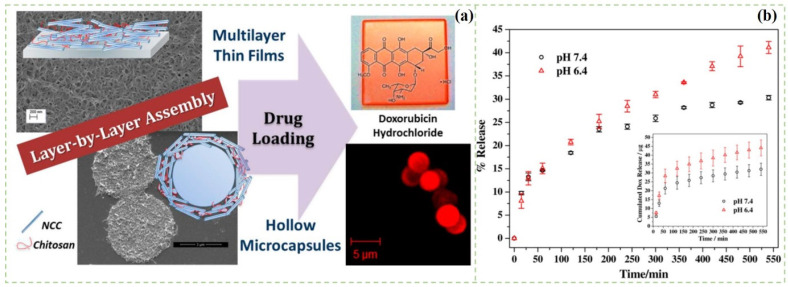
(**a**) Schematic illustration of the preparation of the nanocrystalline/chitosan; (**b**) Release profile of doxorubicin hydrochloride from the composite film at 37 °C using PBS buffer as release media [83].

**Figure 4 polymers-14-02648-f004:**
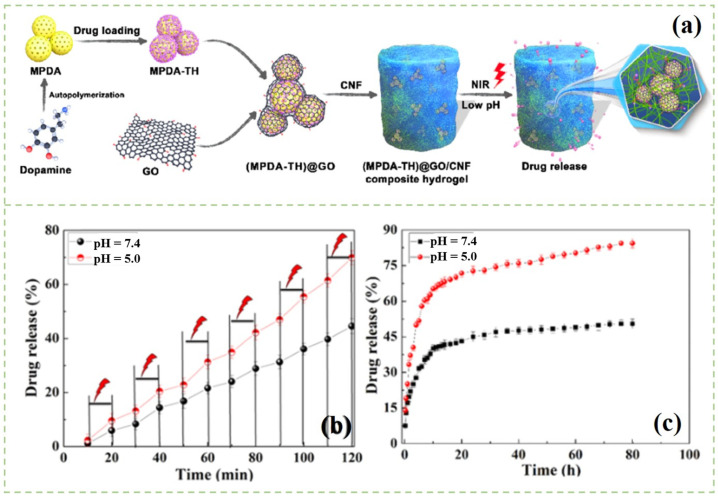
(**a**) Schematic illustration of the fabrication of the MPDA@GO/CNFs composite hydrogel; (**b**) NIR-light-triggered drug release behavior from MPDA@GO/CNFs composite hydrogel in different PBS solution at 37 °C; (**c**) In vitro drug release profiles of the MPDA@GO/CNFs composite hydrogel [106].

**Figure 5 polymers-14-02648-f005:**
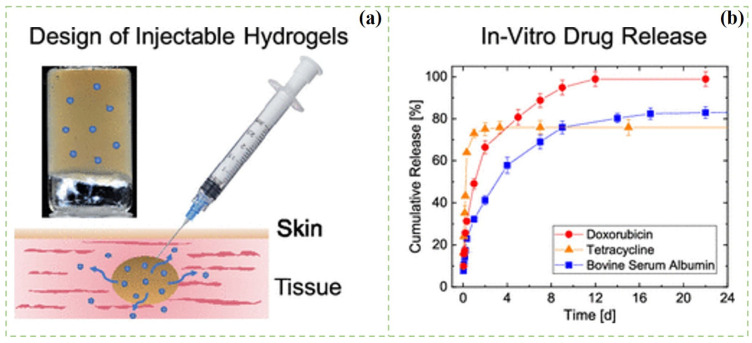
(**a**) Schematic illustration of the preparation of the injectable CNCs hydrogels; (**b**) In vitro drug release from CNCs hydrogels [111].

**Figure 6 polymers-14-02648-f006:**
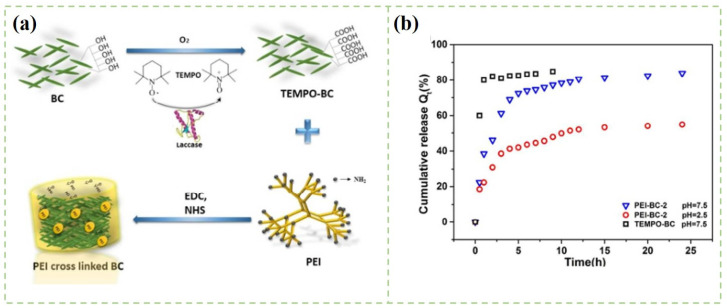
(**a**) Schematic illustration of the preparation of the PEI-BC composite aerogels; (**b**) The cumulative release curves of aspirin [128].

**Figure 7 polymers-14-02648-f007:**
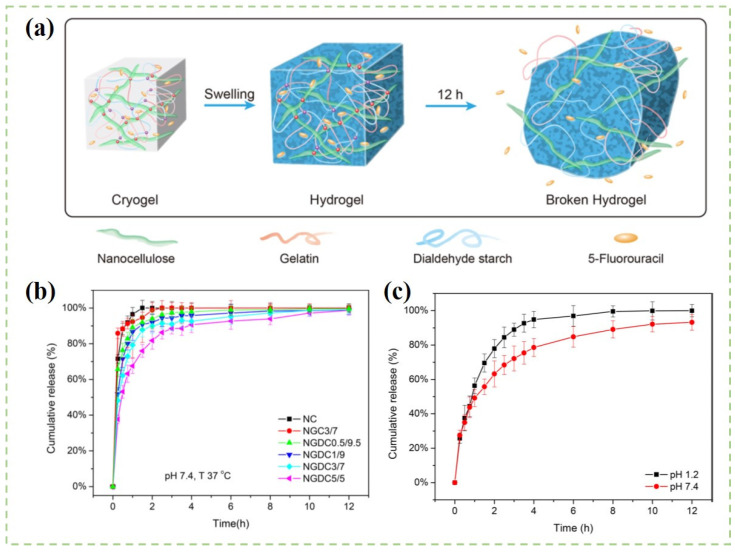
(**a**) Schematic illustration of the preparation of the 5-fluorouracil-loaded CNF/gelation hydrogels; (**b**) Drug release profiles of different hydrogels; (**c**) Drug release profiles of the hydrogel at different pH environments [154].

**Figure 8 polymers-14-02648-f008:**
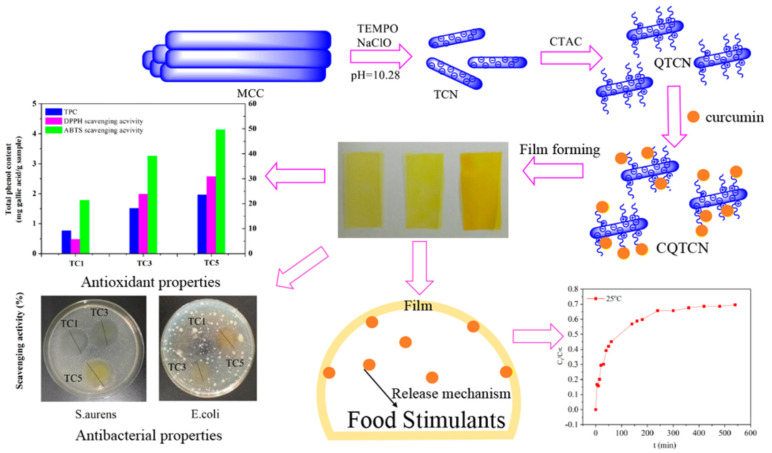
Schematic illustration of curcumin-loaded membrane and the drug release curve of curcumin [167].

**Table 1 polymers-14-02648-t001:** Comparison of structure parameters of different types of nanocellulose.

Types of Nanocellulose	Structure Properties			Mechanical Properties	
	Diameter (nm)	Length (nm)	Crystallinity (%)	Young’s Modulus (GPa)	Tensile Strength (GPa)
CNCs	3–50	100–500	~90	50–140	8–10
CNFs	3–60	≥10^3^	50–90	50–160	0.8–1
BC	20–100	≥10^3^	84–89	78	0.2–2

**Table 2 polymers-14-02648-t002:** Nanocellulose-based materials used in hydrophilic-drug delivery applications.

Dimensions	Drug Delivery Carriers	Hydrophilic-Drug Models	Drug Release Behaviors	Drug Release Mechanism	References
1D	Poly(lactic acid)/CNCs nanocomposite fibers	Columbia blue	Little burst release (<5%) in the first 4 h.	Fickian diffusion	[140]
	CNCs-hordein/ zein fibers	Riboflavin	After 24 h, the cumulative release amount was 26.99%.	-	[141]
	CNFs/poly(N-isopropylacrylamide) hybrid microspheres	5-Fluorouracil	The cumulative drug released amount was 89% within 1 h at 22 °C.	Fickian diffusion	[142]
	CNCs/chitosan particles	Procaine hydrochloride	In the first 10 min, drug release rate was relatively fast; then, it became slower in the next 1 h.	-	[143]
2D	Nanocellulose/ pectin films	Hydroxychloroquine	In the first 2 h, the drug release amount from the pectin films containing CNCs and CNFs was approximately 65% and 95%, respectively.	Fick’s diffusion	[144]
	Chitosan/CNCs films	Doxorubicin	Under acidic pH conditions, the drug release amount is higher.	Fickian diffusion	[83]
	CNFs/polyvinyl alcohol films	Acetaminophen	-	Diffusion controlled and burst release	[145]
	BC composite membranes	Tetracycline hydrochloride	The drug release amount was 90% within 10 h in HEPES buffers.	-	[146]
3D	Polyacrylamide/ CNFs hybrid hydrogels	Niacinamide	The cumulative drug release amount was 45% with 350 min.	Pseudo-Fickian diffusion	[147]
	CNFs/polydopamine composite hydrogels	Tetracycline hydrochloride	In acid PBS solution, 70% of the loaded drugs were released after 15 h.	Anomalous transport	[148]
	CNFs/polyethylenimine aerogels	Sodium salicylate	In SIF condition with a pH of 7.4, the cumulative drug release amount was 78.49%.	Pseudo-second-order release	[130]
	CNFs aerogel	Bendamustine hydrochloride	The cumulative drug release amount was 78% ± 2.28% in 24 h.	Non-Fickian mechanism	[149]
	CNFs/ polyethyleneimine-N/ isopropylacrylamide aerogel	Doxorubicin	The cumulative drug release amount was 59.45% at pH of 3 and 37 °C.	-	[150]
	CNFs/hydroxypropylmethylcellulose nanocomposites	Ketorolac tromethamine	The cumulative drug release amount was 95.12% after 8 h under PBS conditions of 7.4.	Non-Fickian diffusion	[151]
	Mesoporous polydopamine@ graphene oxide/CNFs composite hydrogel	Tetracycline hydrochloride	In the first 1 h, burst release amount was 14% in PBS 7.4 solution. The maximum TH release (84.3%) was achieved in 72 h in PBS 5.0 solution.	Anomalous transport	[106]

**Table 3 polymers-14-02648-t003:** Nanocellulose-based materials used in hydrophobic-drug delivery applications.

Dimensions	Drug Delivery Carriers	Hydrophobic-Drug Models	Drug Release Behaviors	Drug Release Mechanism	References
1D	CNCs–cetyltrimethylammonium bromide suspensions	Paclitaxel, docetaxel, and etoposide	A total of 75% of the drug was released over 4 days.	-	[159]
	CNCs/rarasaponin particles	Tetracycline	More drugs released from neutral condition than in acid condition.	Pseudo-first-order	[160]
2D	BC/hyaluronic acid/ diclofenac films	Diclofenac sodium	The maximum cumulative release was 90% which was obtained after 4 min in simulated salivary fluid.	Non-Fickian transport	[161]
	CNFs/poly(glycerol sebacate)/polypyrrole patches	Curcumin	The cumulative drug-released amount was less than 2% with five months in PBS under pH of 7.4.	-	[162]
	BC/polyvinyl alcohol films	Vanillin	The diffusion process reached equilibrium after 1 h in water.	Fickian diffusion	[163]
3D	BC/sodium alginate hybrid hydrogels	Ibuprofen	During the first 2 h, the drug release amount was less than 10% in acidic conditions with the pH condition of 1.5.	Non-Fickian diffusion	[164]
	Polyethylenimine/BC aerogels	Aspirin	The cumulative drug release was 80.6% with 25 h in pH condition of 7.5.	-	[128]
	Zeolitic imidazolate framework-8@PCNFs composite hydrogel	Curcumin	Under pH condition of 2.5, the maximum curcumin release amount was 90%.	Anomalous transport	[165]

**Table 4 polymers-14-02648-t004:** Exponent *n* and the drug release mechanism from the controlled-drug-delivery carriers of different geometry [148].

Exponent, *n*			Drug Release Mechanisms
Thin Films	Cylinders	Spheres	
0.5	0.45	0.43	Fickian diffusion
0.5 < *n* < 1	0.45 < *n* < 0.89	0.43 < *n* < 0.85	Anomalous transport
1	0.89	0.85	Case-II transport

## Data Availability

Not applicable.

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
