# Peer review of "Nanocellulose-Based Composite Materials Used in Drug Delivery Systems"

_polymers, 2022, doi:10.3390/polym14132648_

Round 1
Reviewer 1 Report
This review focus on nanocellulose based materials for drug delivery, analyzing their preparation, composition, drug release mechanisms as well as their applications. The manuscript is well written, organized and very interesting. I have few minor comments:
- In Table 2 and Table 3, when you are talking about the drug delivery carriers that have been investigated I would add if there are 1D, 2D or 3D structure and group together.
- I would be more precise about the future perspectives at Section 5.
- In the reference section, it would be useful to have the DOI number at the end of each reference.
Author Response
Responses to reviewers’ comments
Manuscript title: Nanocellulose Based Composite Materials Used in Drug Delivery Systems
Manuscript ID: polymers-1755779
Reviewer(s)' Comments to Author:
Reviewer: 1
Comments:
This review focus on nanocellulose based materials for drug delivery, analyzing their preparation, composition, drug release mechanisms as well as their applications. The manuscript is well written, organized and very interesting. I have few minor comments:
Reponses:
We appreciate the reviewer’s positive comments and constructive suggestions. The followings are our detailed responses to the reviewer’s comments.
- In Table 2 and Table 3, when you are talking about the drug delivery carriers that have been investigated I would add if there are 1D, 2D or 3D structure and group together.
Reponses:
Thanks for the suggestion. The information of 1D, 2D or 3D structures towards different drug delivery carriers were added in the Table 2 and Table 3.
- I would be more precise about the future perspectives at Section 5.
Reponses:
Thanks. The future perspectives were added in the revised manuscript in Section 5.
Nanocellulose has been proved to be one of the most prominent green materials in various applications, and it has attracted great interests in academic research and industrial applications, as evidenced by more than 4500 relevant patents and commercial products around the world. This review comprehensively introduced the latest research progress of different nanocellulose based composite in drug delivery. The unique properties of the materials used in drug delivery systems are discussed systematically. As a natural material, nanocellulose has the potential to be involved in future medical applications. The current research is still at the scientific research stage, and we need to focus on the in vivo drug release research on animals in the future. More long-term studies are needed to analyze and assess the potential effects of nanocellulose on humans. Nonetheless, it is difficult for the human body to degrade nanocellulose based materials, and the interaction mechanism between nanocellulose and cells is still unclear. In the future, it is necessary to explore whether the introduction of nanocellulose has potential harmful to the skin, such as causing skin hyperplasia and scar or other complications.
From a scientific and economic point of view, nanocellulose is a resource and gift provided by nature. Driven by the recent extraordinary activities in the field of biomedical applications, nanocellulose will make a breakthrough in drug delivery system. Moreover, with the continuous optimization and commercialization of nanocellulose, nanocellulose based composite materials with better-designed structure and multi-functions (e.g. pH or NIR responses) will be undoubtedly extensively used in the field of drug delivery. Despite nanocellulose-based composites have already shown great promise in biomedical fields, the large-scale commercial applications of nanocellulose are closely related to the structure and performance of the materials. So, we need commit to solving the current difficulties and challenges. Therefore, with the continuous optimization of nanocellulose production, modification, industrialization, and commercialization, nanocellulose will undoubtedly shine in more fields.
- In the reference section, it would be useful to have the DOI number at the end of each reference.
Reponses:
Thanks for the suggestion. The DOI number of each reference was added in the reference section.
As last, we deeply appreciated the great efforts and suggestions from the reviewers. Their great contribution much improved the quality of our manuscript.
Reviewer 2 Report
Dear author, in manuscript Nanocellulose Based Composite Materials Used in Drug Delivery Systems, summarized a very interesting research based facts and information it will be very helpful to the researchers, I would like to suggest some pints please revise your manuscript to the following suggested points, I strongly recommend revising this manuscript as follows:
1. Author should add some information about the crosslinked biomaterial based materials reported for sustained release application, those also have properties such as biodegradability and biocompatibility with tunable physicochemical properties. The author should refer to and cite the following articles in the revised manuscript. 1. Effect of Polyethylene Glycol on Properties and Drug Encapsulation–Release Performance of Biodegradable/Cytocompatible Agarose–Polyethylene Glycol–Polycaprolactone Amphiphilic Co-Network Gels and 2. Degradable/cytocompatible and pH-responsive amphiphilic conetwork gels based on agarose-graft copolymers and polycaprolactone 3. Reactive compatibilizer mediated precise synthesis and application of stimuli-responsive polysaccharides-polycaprolactone amphiphilic co-network gels.
2. Author should add the information about the commercially available biomaterials based drug delivery products in the revised manuscript.
3. I would like the suggest to authors please add a separate paragraphs for a future perspective with global market value and growth rate of suck materials.
4. A very new interesting chemistry( Nucleophilic substitution) based crosslinked biomaterials based film and hydrogels have been reported recently with numerous advantageous properties for drug delivery and tissue engineering application. The author should refer in the revised manuscript to the following articles 1. Dually crosslinked injectable hydrogels of poly (ethylene glycol) and poly [(2-dimethylamino) ethyl methacrylate]-b-poly (N-isopropyl acrylamide) as a wound healing promoter. 2. Self-Assembly of Partially Alkylated Dextran-graft-poly[(2-dimethylamino)ethyl methacrylate] Copolymer Facilitating Hydrophobic/Hydrophilic Drug Delivery and Improving Conetwork Hydrogel Properties.3 Liquid prepolymer-based in situ formation of degradable poly (ethylene glycol)-linked-poly (caprolactone)-linked-poly (2-dimethylaminoethyl) methacrylate amphiphilic conetwork gels showing polarity driven gelation and bioadhesion." ACS Applied Bio Materials 1, no. 5 (2018): 1606-1619. 4. Synthesis and tailoring the degradation of multi-responsive amphiphilic conetwork gels and hydrogels of poly (β-amino ester) and poly (amido amine). Polymer, 111, 265-274.
5. Cellulose based sheet materials with antibacterial and wound healing properties have been reported recently author should cite in the revised manuscript in the appropriate place. Silver-loaded carboxymethyl cellulose nonwoven sheet with controlled counterions for infected wound healing." Carbohydrate Polymers 286 (2022): 119289.
6. Author should add the chemical structures of biopolymers and their modification that will help to understand the readers cite the following articles for it. Advancement of Biomaterial‐Based Postoperative Adhesion Barriers." Macromolecular bioscience 21, no. 3 (2021): 2000395.
Author Response
Responses to reviewers’ comments
Manuscript title: Nanocellulose Based Composite Materials Used in Drug Delivery Systems
Manuscript ID: polymers-1755779
Reviewer(s)' Comments to Author:
Reviewer: 2
Comments:
Dear author, in manuscript Nanocellulose Based Composite Materials Used in Drug Delivery Systems, summarized a very interesting research based facts and information it will be very helpful to the researchers, I would like to suggest some pints please revise your manuscript to the following suggested points, I strongly recommend revising this manuscript as follows:
Reponses:
We appreciate the reviewer’s constructive comments. The following is our detailed responses point by point.
- Author should add some information about the crosslinked biomaterial based materials reported for sustained release application, those also have properties such as biodegradability and biocompatibility with tunable physicochemical properties. The author should refer to and cite the following articles in the revised manuscript. 1. Effect of Polyethylene Glycol on Properties and Drug Encapsulation-Release Performance of Biodegradable/Cytocompatible Agarose-Polyethylene Glycol-Polycaprolactone Amphiphilic Co-Network Gels and 2. Degradable/cytocompatible and pH-responsive amphiphilic conetwork gels based on agarose-graft copolymers and polycaprolactone 3. Reactive compatibilizer mediated precise synthesis and application of stimuli-responsive polysaccharides-polycaprolactone amphiphilic co-network gels.
Page 7, lines 264-266. Crosslinked biomaterials for drug delivery commonly need to have biodegradability, biocompatibility, and adjustable physicochemical properties [91-93].
The related references were added in the revised manuscript.
[91] Chandel, A.K.; Kumar, C.U.; Jewrajka, S.K. Effect of polyethylene glycol on properties and drug encapsulation-release performance of biodegradable/cytocompatible agarose-polyethylene glycol-polycaprolactone amphiphilic co-network gels. ACS Appl. Mater. Interfaces 2016, 10, 3182-3192. https://doi.org/10.1021/acsami.5b10675.
[92] Bera, A; Singh, A.K.; Uday Kumar, C.; Jewrajka, S.K. Degradable/cytocompatible and pH responsive amphiphilic conetwork gels based on agarose-graft copolymers and polycaprolactone. J Mater Chem B 2015, 21, 8548-8557. https://doi.org/10.1039/c5tb01251a.
[93] Singh, A.K.; Anupam, B.; Bhingaradiya, N.; Jewrajka, S.K. Reactive compatibilizer mediated precise synthesis and application of stimuli responsive polysaccharides-polycaprola-ctone amphiphilic co-network gels, Polymer 2016, 99, 470-479. https://doi.org/10.1016/j.polymer.2016.07.033.
- Author should add the information about the commercially available biomaterials based drug delivery products in the revised manuscript.
Reponses:
Thanks for the suggestion.
Page 1, lines 33-36, Commercially available poly(lactide-co-glycolide) (PLGA) based materials are often used for particle drug release formulations [4]. However, due to its large burst release and acidic degradation behaviors, they are often limited to a certain extent.
The related reference was added in the revised manuscript.
[4] Miles, C.E.; Gwin, C.; Zubris, K. Tyrosol derived poly(ester-arylate)s for sustained drug delivery from microparticles. ACS Biomater. Sci. Eng. 2021, 7, 2580-2591. https://doi.org/10.1021/acsbiomaterials.1c00448.
- I would like the suggest to authors please add a separate paragraphs for a future perspective with global market value and growth rate of suck materials.
Reponses:
Thanks for the suggestion. The paragraphs were added in the Section 5.
Nanocellulose has been proved to be one of the most prominent green materials in various applications, and it has attracted great interests in academic research and industrial applications, as evidenced by more than 4500 relevant patents and commercial products around the world. This review comprehensively introduced the latest research progress of different nanocellulose based composite in drug delivery. The unique properties of the materials used in drug delivery systems are discussed systematically. As a natural material, nanocellulose has the potential to be involved in future medical applications. The current research is still at the scientific research stage, and we need to focus on the in vivo drug release research on animals in the future. More long-term studies are needed to analyze and assess the potential effects of nanocellulose on humans. Nonetheless, it is difficult for the human body to degrade nanocellulose based materials, and the interaction mechanism between nanocellulose and cells is still unclear. In the future, it is necessary to explore whether the introduction of nanocellulose has potential harmful to the skin, such as causing skin hyperplasia and scar or other complications.
From a scientific and economic point of view, nanocellulose is a resource and gift provided by nature. Driven by the recent extraordinary activities in the field of biomedical applications, nanocellulose will make a breakthrough in drug delivery system. Moreover, with the continuous optimization and commercialization of nanocellulose, nanocellulose based composite materials with better-designed structure and multi-functions (e.g. pH or NIR responses) will be undoubtedly extensively used in the field of drug delivery. Despite nanocellulose-based composites have already shown great promise in biomedical fields, the large-scale commercial applications of nanocellulose are closely related to the structure and performance of the materials. So, we need commit to solving the current difficulties and challenges. Therefore, with the continuous optimization of nanocellulose production, modification, industrialization, and commercialization, nanocellulose will undoubtedly shine in more fields.
- A very new interesting chemistry (Nucleophilic substitution) based crosslinked biomaterials based film and hydrogels have been reported recently with numerous advantageous properties for drug delivery and tissue engineering application. The author should refer in the revised manuscript to the following articles 1. Dually crosslinked injectable hydrogels of poly (ethylene glycol) and poly [(2-dimethylamino) ethyl methacrylate]-b-poly (N-isopropyl acrylamide) as a wound healing promoter. 2. Self-Assembly of Partially Alkylated Dextran-graft-poly[(2-dimethylamino)ethyl methacrylate] Copolymer Facilitating Hydrophobic/Hydrophilic Drug Delivery and Improving Conetwork Hydrogel Properties.3 Liquid prepolymer-based in situ formation of degradable poly (ethylene glycol)-linked-poly (caprolactone)-linked-poly (2-dimethylaminoethyl) methacrylate amphiphilic conetwork gels showing polarity driven gelation and bioadhesion." ACS Applied Bio Materials 1, no. 5 (2018): 1606-1619. 4. Synthesis and tailoring the degradation of multi-responsive amphiphilic conetwork gels and hydrogels of poly (β-amino ester) and poly (amido amine). Polymer, 111, 265-274.
Reponses:
Thanks for the suggestion.
Page 9, lines 313-314. Nucleophilic substitution method can be used to prepare injectable hydrogels for drug delivery and tissue engineering applications [107-110].
The related references were added in the revised manuscript.
[107] Chandel, A K S.; Kannan, D.; Nutan, B.; Singh, S.; Jewrajka, S K. Dually crosslinked injectable hydrogels of poly (ethylene glycol) and poly [(2-dimethylamino) ethyl methacrylate]-b-poly (N-isopropyl acrylamide) as a wound healing promoter. J Mater Chem B 2017, 5(25), 4955-4965. https://doi.org/10.1039/C7TB00848A.
[108] Chandel, A K S.; Nutan, B.; Raval, I H.; Jewrajka, S K. Self-assembly of partially alkylated dextran-graft-poly [(2-dimethylamino) ethyl methacrylate] copolymer facilitating hydrophobic/hydrophilic drug delivery and improving conetwork hydrogel properties. Biomacromolecules 2018, 19(4), 1142-1153. https://doi.org/10.1021/acs.biomac.8b00015.
[109] Nutan, B.; Chandel, A K S.; Jewrajka, S K. Liquid prepolymer-based in situ formation of degradable poly (ethylene gly-col)-linked-poly (caprolactone)-linked-poly (2-dimethylaminoethyl) methacrylate amphiphilic conetwork gels showing po-larity driven gelation and bioadhesion. ACS Applied Bio Materials 2018, 1(5), 1606-1619. https://doi.org/10.1021/acsabm.8b00461
[110] Nutan, B.; Chandel, A K S.; Bhalani, D V.; Jewrajka, S K. Synthesis and tailoring the degradation of multi-responsive am-phiphilic conetwork gels and hydrogels of poly (β-amino ester) and poly (amido amine). Polymer 2017, 111, 265-274. https://doi.org/10.1016/j.polymer.2017.01.057.
- Cellulose based sheet materials with antibacterial and wound healing properties have been reported recently author should cite in the revised manuscript in the appropriate place. Silver-loaded carboxymethyl cellulose nonwoven sheet with controlled counterions for infected wound healing." Carbohydrate Polymers 286 (2022): 119289.
Reponses:
Thanks. As suggested, the corresponding information was added in revision.
Page 13, lines 418-419. Cellulose based sheet materials with antibacterial and wound healing properties have been reported [152].
The related references were added in the revised manuscript.
[152] Ohta, S.; Mitsuhashi, K.; Chandel, A K S.; Qi, P.; Nakamura, N.; Nakamichi, A.; Yoshida, H.; Yamaguchi, G.; Hara, Y.; Sasaki, R. Silver-loaded carboxymethyl cellulose nonwoven sheet with controlled counterions for infected wound healing. Carbohyd Polym 2022, 286, 119289. https://doi.org/10.1016/j.carbpol.2022.119289.
- Author should add the chemical structures of biopolymers and their modification that will help to understand the readers cite the following articles for it. Advancement of Biomaterial‐Based Postoperative Adhesion Barriers." Macromolecular bioscience 21, no. 3 (2021): 2000395.
Reponses:
Thanks for your kind suggestions.
This review was mainly related to the nanocellulose based materials for drug delivery applications. The above review was based on natural and synthetic biomaterials such as alginate, hyaluronan, cellulose starch, chondroitin sulfate, polyethylene glycol, polylactic acid. The chemical structures and their modification were different from nanocellulose. Thus, the chemical structures of biopolymers and their modification were not added in the review. The paper was cited in the references part.
The reference was added in the revised manuscript.
[46] Chandel, A K S.; Shimizu, A.; Hasegawa, K.; Ito, T. Advancement of biomaterial‐based postoperative adhesion barriers. Macromol biosci 2021, 21(3), 2000395. https://doi.org/10.1002/mabi.202000395.
As last, we deeply appreciated the great efforts and suggestions from the reviewers. Their great contribution much improved the quality of our manuscript.
Reviewer 3 Report
Please see the attachment.

Author Response
Responses to reviewers’ comments
Manuscript title: Nanocellulose Based Composite Materials Used in Drug Delivery Systems
Manuscript ID: polymers-1755779
Reviewer(s)' Comments to Author:
Reviewer: 3
Comments:
In this manuscript, the authors reviewed the recent progress of nanocellulose-based composite materials for drug delivery systems. The structure and logic of this review are well, but the authors need to address several major issues in this manuscript before publication in Polymers.
Reponses:
We appreciate the reviewer’s valuable suggestions and comments. The detailed responses to the reviewer’s comments are listed below.
- There are already several reviews published in 2021 and 2022 about the same topic [1-4]. The authors need to elaborate on why this review paper is still important and what is the major difference that distinguishes this review from the published ones.
[1] Raghav, N., Sharma, M. R., & Kennedy, J. F. (2021). Nanocellulose: A mini-review on types and use in drug delivery systems. Carbohydrate Polymer Technologies and Applications, 2, 100031.
[2] Das, S., Ghosh, B., & Sarkar, K. (2022). Nanocellulose as sustainable biomaterials for drug delivery. Sensors International, 3, 100135.
[3] Patil, T. V., Patel, D. K., Dutta, S. D., Ganguly, K., Santra, T. S., & Lim, K. T. (2022). Nanocellulose, a versatile platform: From the delivery of active molecules to tissue engineering applications. Bioactive materials, 9, 566-589.
[4] Shi, Y., Jiao, H., Sun, J., Lu, X., Yu, S., Cheng, L., Liu, J. (2022). Functionalization of nanocellulose applied with biological molecules for biomedical application: A review. Carbohydrate Polymers, 119208.
Reponses:
Thanks for the suggestion.
Page 2, lines 54-61. This review focuses on the functions of nanocellulose based composite materials with different dimensional (1D, 2D, 3D) used in drug delivery systems. Hydrophilic and hydrophobic drug release behaviors of nanocellulose based materials are systematic summarized for the first time. The relationships between the structures of nanocellulose based materials and drug release behaviors are also emphasized. Moreover, the latest research work of nanocellulose based materials used in drug delivery is introduced in a general overview. The future perspectives with global market value and growth rate of nanocellulose materials are also systematic summarized.
Therefore, this review is distinctly different from above mentioned reviews.
- There are several places where the authors’ statements lack accuracy. For example, in line 116, “It is reported that CNFs prepared by enzymatic hydrolysis has no cytotoxicity at any concentration”, while Ref. 48 presents a specific concentration. Another example is line 140, “Cellulose, non-synthetic or redeveloped cellulose with hydrophilicity can decompose rapidly at ambient temperature”. Do authors indicate that cellulose can spontaneously decompose at room temperature in the air? The authors do need to re-check the manuscript elsewhere to improve the accuracy of their statements.
Reponses:
Thank you for pointing it out. Sorry for misunderstanding. The information was modified in the revised manuscript.
Page 4, Lines 123-124, It is reported that CNFs prepared by enzymatic hydrolysis has no cytotoxicity at tested concentrations (~10-1000 μg/mL) [49].
Page 4, Lines 158-161, It is generally believed that cellulose does undergo chemical decomposition due to an elevated temperature. One of the main volatile decomposition products is levoglucosan (LGA). This process usually leaves a solid carbon residue whose chemical and physical composition are mostly unknown [63].
The related references were added in the revised manuscript.
[49] Kumari, P.; Pathak, G.; Gupta, R.; Sharma, D.; Meena, A. Cellulose nanofibers from lignocellulosic biomass of lemongrass using enzymatic hydrolysis: characterization and cytotoxicity assessment. Daru 2019, 27, 683-693. https://doi.org/10.1007/s40199-019-00303-1.
[63] Paajanen, A.; Vaari, J. High-temperature decomposition of the cellulose molecule: a stochastic molecular dynamics study. Cellulose 2017, 24(7):2713-2725. https://doi.org/10.1007/s10570-017-1325-7.
- The authors narrowed the biocompatibility to toxicity to cells/tissues. However, it is important to include immunogenicity into consideration. Would nanocellulose-based materials trigger the immune response and induce inflammation? What is the mechanism for pro-inflammatory or anti-inflammatory responses?
Reponses:
Many thanks for the constructive suggestions.
Page 4, lines 141-151. Moreover, the biocompatibility of nanocellulose depends on its structural characteristics, application concentration, research model, cells type and exposure time. The uptake of nanocellulose into cells is usually low, which will not induce oxidative stress, and will not produce obvious cytotoxic and genotoxic effects. However, macrophages can internalize rod-shaped CNCs due to their phagocytic function, which can lead to moderate to severe inflammatory response. The response is mainly depended on the functionalization of CNCs [58]. By introducing different chemical groups on the surface of nanocellulose, the pro-inflammatory response of nanocellulose can be significantly reduced [59]. Therefore, it is necessary to conduct additional immunological studies on nanocellulose based materials to better understand its impact on innate and adaptive immunity. However, compared with other materials, nanocellulose based materials are still preferable because their cytotoxicity is relatively low.
The related references were added in the revised manuscript.
[58] Célia, Ventura.; Fátima, Pinto.; Loureno, A.F. On the toxicity of cellulose nanocrystals and nanofibrils in animal and cellular models. Cellulose 2020, 27(10). https://doi.org/10.1007/s10570-020-03176-9.
[59] Čolić M, Tomić S, Bekić M. Immunological aspects of nanocellulose. Immunol Lett 2020, 222, 80-89. https://doi.org/10.1016/j.imlet.2020.04.004.
- Biodegradability is the capacity for biological degradation of organic materials by living organisms down to the base substances such as water, carbon dioxide, methane, basic elements, and biomass [5]. The authors claimed that nanocellulose-based composite materials are biodegradable in vivo. The mechanism behind biodegradability needs to be elaborated. Please be aware that absorption and removal of nanocellulose-based materials may not be considered biodegradability as they are physically removed instead of chemically degraded.
[5] Goswami, P., & O'Haire, T. (2016). Developments in the use of green (biodegradable), recycled and biopolymer materials in technical nonwovens. Advances in Technical Nonwovens, 97-114.
Reponses:
Thank you for pointing it out.
Page 4, lines 162-166. However, the biodegradability of nanocellulose in animal and human tissues does not clear, since cells are not able to synthesize cellulases. Nonenzymatic, spontaneous biodegradability of cellulose chains may perhaps explain the slow breakdown of unaltered cellulose within the human body [65]. But this is admittedly conjecture and it has not been adequately studied [65].
The related reference was added in the revised manuscript.
[65] Czaja, W.K.; Young, D.J.; Kawecki, M.; Browm, R.M. The future prospects of microbial cellulose in biomedical applications. Biomacromolecules 2007, 8, 1-12. https://doi.org/10.1021/bm060620d.
- The authors limited drug delivery to small molecule delivery. Delivery of proteins and nucleic acids should be included as well.
Reponses:
Thanks for the suggestion.
Page 11, lines 405-414. Despite to deliver the small drug molecule as listed in Table 2, nanocellulose based materials can be used to deliver proteins and nucleic acids. Because nanocellulose based materials can meet the strict medical requirements of appropriate carriers for protein and nucleic acid fixation. Basu et al. developed calcium chloride crosslinked CNFs hydrogels for transporting biomolecules [139]. Bovine serum albumin protein was loaded into hydrogel by simple immersion method. The large positively charged proteins promote the sustained drug release behavior of CNFs. The positively charged proteins also increased the mechanical strength of the composite hydrogels. The electrostatic interaction between protein and hydrogel was the main factor to promote the physical adsorption of hydrogel structure stability and activity. Therefore, calcium crosslinked CNFs hydrogels can transport proteins without affecting their activity.
The related reference was added in the revised manuscript.
[139] Alex, B.; Maria, S.; Natalia, F. Towards tunable protein-carrier wound dressings based on nanocellulose hydrogels cross-linked with calcium ions. Nanomaterials 2018, 550. https://doi.org/10.3390/nano8070550.
- The manuscript included several figures adapted from previous publications. However, no claims of copyright permission from the publishers were found.
Reponses:
Thanks. The copyright permission was obtained from the publishers and it was uploaded together with the revised manuscript.
As last, we deeply appreciated the great efforts and suggestions from the reviewers . Their great contribution much improved the quality of our manuscript.
Round 2
Reviewer 3 Report
In this revised manuscript, the authors have addressed all my previous comments. Therefore, I recommend acceptance of this revised manuscript.
Author Response
Comments and Suggestions for Authors:
In this revised manuscript, the authors have addressed all my previous comments. Therefore, I recommend acceptance of this revised manuscript.
Reponses: Thanks. We appreciate the reviewer’s approval of our manuscripts.